# Enhancing Mechanical and Corrosion Properties of AISI 420 with Titanium-Nitride Reinforcement through High-Power-Density Selective Laser Melting Using Two-Stage Mixed TiN/AISI 420 Powder

**DOI:** 10.3390/ma16114198

**Published:** 2023-06-05

**Authors:** Duc Tran, Chih-Kuang Lin, Pi-Cheng Tung, Jeng-Rong Ho, Jason Shian-Ching Jang, Jing-Chie Lin, I-Yu Tsao, Thanh-Long Le

**Affiliations:** 1Department of Mechanical Engineering, National Central University, Jhong-Li District, Tao-Yuan City 32001, Taiwan; tranducoto@hcmut.edu.vn (D.T.); t330014@cc.ncu.edu.tw (C.-K.L.); t331166@ncu.edu.tw (P.-C.T.); 2Institute of Material Science and Engineering, National Central University, Jhong-Li District, Tao-Yuan City 32001, Taiwan; jscjang@ncu.edu.tw (J.S.-C.J.); jclincom@cc.ncu.edu.tw (J.-C.L.);; 3Faculty of Mechanical Engineering, Ho Chi Minh City University of Technology, Ho Chi Minh City 700000, Vietnam; ltlong@hcmut.edu.vn

**Keywords:** two-stage powder mixing scheme, selective laser melting, TiN/AISI 420 composites, microstructure, corrosion resistance

## Abstract

This study investigates the effect of laser volume energy density (VED) on the properties of AISI 420 stainless steel and TiN/AISI 420 composite manufactured by selective laser melting (SLM). The composite contained 1 wt.% TiN and the average diameters of AISI 420 and TiN powders were 45 µm and 1 µm, respectively. The powder for SLMing the TiN/AISI 420 composite was prepared using a novel two-stage mixing scheme. The morphology, mechanical, and corrosion properties of the specimens were analyzed, and their correlations with microstructures were investigated. The results showed that the surface roughness of both SLM samples decreases with increasing VED, while relative densities greater than 99% were achieved at VEDs higher than 160 J/mm^3^. The SLM AISI 420 specimen fabricated at a VED of 205 J/mm^3^ exhibited the highest density of 7.7 g/cm^3^, tensile strength (UTS) of 1270 MPa, and elongation of 3.86%. The SLM TiN/AISI 420 specimen at a VED of 285 J/mm^3^ had a density of 7.67 g/cm^3^, UTS of 1482 MPa, and elongation of 2.72%. The microstructure of the SLM TiN/AISI 420 composite displayed a ring-like micro-grain structure consisting of retained austenite on the grain boundary and martensite in the grain. The TiN particles strengthened the mechanical properties of the composite by accumulating along the grain boundary. The mean hardnesses of the SLM AISI 420 and TiN/AISI 420 specimens were 635 and 735 HV, respectively, which exceeded previously reported results. The SLM TiN/AISI 420 composite exhibited excellent corrosion resistance in both 3.5 wt.% NaCl and 6 wt.% FeCl_3_ solutions, with a resulting corrosion rate as low as 11 µm/year.

## 1. Introduction

Manufacturing using injection molds is essential for high-volume production, but there are challenges in the manufacturing molds, such as failure due to high loads and surface corrosion due to high temperature and pressure after long-term use. As a result, reusing injection molds can have a detrimental effect on the quality of injection products, further reducing their lifespan. Moreover, to enhance mold productivity, complex cooling channels are required compared to traditional molds with a single cooling channel [1]. Hence, identifying appropriate materials and production methods for molds is essential. 

Selective laser melting (SLM) is an additive manufacturing technique that involves melting, shaping, and consolidating feedstock layer by layer using a computer-controlled laser beam [2]. Unlike traditional manufacturing methods, SLM provides a fast and efficient process for producing finished parts without molds, reduces material wastage, eliminates postprocessing, and shortens production lead times [3]. Currently, commercial SLM products made of aluminum alloys, titanium alloys, and stainless steels have been widely used in various fields, such as military, aerospace, and different industrial areas, especially in injection mold manufacturing [4]. Moreover, martensitic stainless steel AISI 420, which contains a high amount of chromium (12–14%) and low carbon (<0.4%), has outstanding mechanical properties and corrosion resistance, making it a suitable material for mold production [5]. Consequently, selective laser melting of AISI 420 for mold manufacturing has attracted much attention.

Various solutions are available to address the mold manufacturing issues mentioned above, including developing SLM processing parameters and exploring new AISI-420-based materials. Many studies have been conducted to improve the material properties of SLM-produced AISI 420, which are suitable for mold manufacturing. For instance, Zhao et al. [6] reported that SLM-produced AISI 420 with an energy density of 170 J/mm^3^ had a hardness of 50.7 (Rockwell hardness—HRC), indicating its potential for plastic injection molding. Nath et al. [7] demonstrated that AISI 420 samples produced with a volume energy density (*VED*) of 63 J/mm^3^, achieved a hardness of up to 55 HRC, and a corrosion rate of 28 μm/per year. Similarly, Saeidi et al. [8] achieved a high sample density of 99.8% for SLM-produced AISI 420 with a *VED* of 139 J/mm^3^, resulting in increased strength of 1670 MPa and hardness of up to 650 Vicker Hardness (HV). In addition, Liverani et al. [9] reported that SLM-produced AISI 420 had slightly higher tensile strength and hardness values compared to previous studies, with 1697 MPa and 688 HV, respectively, and exhibited improved hardness and corrosion resistance compared to traditional methods. Although SLM-produced AISI 420 can achieve desirable properties, the material’s complex chemical composition, along with the high cooling and solidification rates of the SLM process, may result in instability in optimization. Therefore, further research on SLM-produced AISI 420 is necessary to address this knowledge gap and ensure effective application in mold manufacturing.

Moreover, to meet the increasing demand for high-performance materials, one promising solution is to reinforce the metallic matrix to enhance its mechanical and chemical properties. Various methods have been developed to reinforce metal materials, including alloying strengthening, precipitation strengthening, metal matrix composite reinforcing, and solid solutions [10]. Among these methods, metal matrix composites have gained significant attention due to their enhanced hardness, excellent wear, and corrosion resistance. Combining the toughness and ductility of metal with the high hardness, stiffness, and wear resistance of ceramics, metal matrix composites create a new material with desirable properties [11].

Currently, the selective laser melting technique has successfully manipulated metallic matrixes with various reinforced particles, including SiC [12], TiB_2_ [13], TiC [14], B_4_C [15], Al_2_O_3_ [16], and TiN [17]. Titanium Nitride (TiN) is an excellent ceramic material with high thermal stability, hardness, wear resistance, and low friction coefficient. TiN is commonly used as a coating material on machine tools, and it has also gained significant attention as a reinforced phase when introduced using the technique of SLM. Studies have demonstrated that TiN can significantly increase the hardness of metal matrix composites [18,19,20]. Considering the advantages of TiN and the excellent results obtained with SLM AISI 420, it is crucial to further investigate the characteristics of SLMed TiN/AISI 420.

From the available information, it appears that there is only one known study on SLM TiN-reinforced AISI 420, which was conducted by Zhao et al. [21]. The study aimed to investigate the effect of different weight percentages of TiN (1%, 3%, and 5%) on the hardness and wear resistance of the produced SLM samples. It found that the SLM TiN/AISI 420 samples exhibited a high hardness of 56.7 HRC when built using a *VED* of 156 J/mm^3^, which was higher than SLM AISI 420s. Additionally, the study also observed a significant improvement in the wear resistance of the TiN-reinforced sample. However, it is essential to note that this study did not address the SLM composite’s strength and corrosion resistance, which are critical properties for molds used in forming plastic, metal, food, and medical components. Additionally, the existing studies on SLM AISI 420 were primarily conducted at relatively low *VED* levels (<200 J/mm^3^), which might result in the presence of more pores or voids in the resulting samples due to insufficient laser processing energy. Therefore, it is worthwhile to conduct comprehensive research to assess the properties of the SLM prints of AISI 420 and TiN/AISI 420, specifically built at high *VED* levels, to meet different application requirements.

The approach of SLM has demonstrated its promising potential for manufacturing high-quality composite workpieces. However, in addition to employing appropriate processing parameters for printing, the quality of the powder used in the SLM process also plays a significant role in the resulting product. Factors such as elemental composition, morphology, flowability, and uniformity in mixing all impact the final product’s quality. Improper preparation methods can lead to poor mixing of composite powders, resulting in inconsistent aggregates or agglomeration of small particles during the printing process, ultimately leading to suboptimal SLM finished products. One prevalent technique to mix SLM powders is ball milling [22], which is cost-effective and can improve powder flowability. However, it also has some drawbacks. For instance, ball milling can result in a nonuniform particle size distribution, which can induce particle agglomeration in the final parts. 

In this study, we utilized a new two-stage mixing method to prepare spherical AISI 420 powders coated with TiN powders and investigated a wider range of *VED* levels, ranging from 130 to 360 J/mm^3^, in the SLM process. We evaluated the physical, mechanical, and corrosion properties of both SLM AISI 420 and 1% TiN/AISI 420 products, as well as examined the microstructure evolutions and mechanisms of enhancing mechanical and corrosion properties caused by the TiN reinforcement particles in the AISI 420 matrix. The findings of this study should be helpful in establishing a standard for the widespread use of TiN/AISI 420 in molding and mold insert manufacturing, providing valuable insights into the development and optimization of SLM composite materials.

## 2. Experimental

### 2.1. Powder Preparation and Selective Laser Melting 

This study aims to create SLM AISI 420 and TiN/AISI 420 composite samples for physical, mechanical, and chemical characterization. To accomplish this, gas-atomized AISI 420 powder (near-spherical in shape, 10–45 µm in size, supplied by Sanyo Co., Ltd., Fujioka, Japan) and Titanium Nitride powder—TiN (polygonal in shape, 1–2 µm in size including nanoparticles, provided by Wellion Co., Ltd., Taichung, Taiwan) were utilized. The chemical compositions of both powders are presented in Table 1.

The proposed method for preparing the mixed TiN and AISI 420 powders with 1 wt% of TiN involves two stages. Stage 1, described in Figure 1, is a two-step mixing process. In Step 1, Figure 1a, the polygonal TiN powder was slowly poured into a beaker containing hexane (C_6_H_12_) solvent, which was placed on an ultrasonic vibrator. Then, the AISI 420 powder was added to the beaker up to 99 wt.%. The powder in the solvent was mixed using a propeller mixer rotating at 600 rpm. At the same time, the beaker was subjected to ultrasonic vibration at a frequency of 40 kHz in continuous mode for 30 min. In Step 2, Figure 1b, the wet mixture was transferred into a bottle and heated at 100 °C to vaporize the retained hexane solvent. The hexane vapor was sucked by a low-pressure vacuum pump and flowed through a low-temperature zone generated by liquid nitrogen, where it was condensed into hexane liquid for recovery. This process operated under a vacuum line system.

By following the aforementioned steps, fine TiN particles were evenly coated onto the surfaces of the AISI 420 powder, while larger TiN particles were uniformly dispersed throughout the AISI 420 powder. The morphologies of the initial and mixed powders are illustrated in Figure 2.

In Stage 2, the TiN/AISI 420 mixed powder was reheated in an oven at 100 °C for 12 h, followed by the removal of particles with a diameter larger than 50 particles through sieving. The as-purchased AISI 420 powder for SLM AISI stainless steel only underwent the final reheating and sieving step. The flow rates of the prepared AISI 420 and TiN/AISI 420 powders were measured using a Hall flowmeter, which determined flow rates of 20 s and 29 s per 50 g, respectively. These flow rates were deemed suitable for the SLM process.

All specimens in this study were printed using the AMP-160 (SLM printer, Tong Tai Machine & Tool Co., Ltd., Kaohsiung, Taiwan), which was equipped with a 500 W fiber laser (IPG), operating at the wavelength of 1070 nm. The Materialise Magics 23.1 software (Leuven, Belgium) was used to plan the sample printing steps and execution of SLM printing parameters. The spot size of the sintering laser beam from the Galvanometer in this machine was 50 µm. To prevent material oxidation during the high-temperature SLM process, this machine was equipped with a nitrogen generator and an oxygen detection sensor to maintain the oxygen levels in the sintering chamber below 600 ppm. The substrate employed was the wrought tool steel (S45C) plate.

To enhance material homogeneity and minimize residual stress in the printed workpiece, we employed the following scanning scheme. As shown in Figure 3a, each layer consisted of mostly rectangular strip printing areas, with each strip composed of interlacing back-and-forth line printing paths. The length of each printing line was 2 mm, and the distance between adjacent lines (also known as the hatch) was fixed at 70 μm. Furthermore, each new printing layer was rotated counterclockwise by 67 degrees relative to the previously printed layer. As depicted in Figure 3b, all samples for various characterizations conducted in this study were printed in a cuboid shape with dimensions of 10 mm × 10 mm × 5 mm in the x, y, and z directions, respectively. Each layer was printed on the X-Y plane, and successive layers were built up along the z direction. Furthermore, Figure 3b displays the dimensions and shape of the tensile test specimen used in the study.

The samples were fabricated using various laser volume energy density (J/mm^3^), *VED*, expressed as $VED=\frac{P}{v\cdot h\cdot t}$ [23], where *P* is the laser power (*W*), *v* is the laser scanning speed (mm/s), *h* is the hatch distance (mm), and *t* is the layer thickness (mm). The employed *VED* in this study ranged from 130 to 360 J/mm^3^. The SLM processing parameters and the sample identification number used in this study are listed in Table 2.

### 2.2. Physical Properties and Microstructure

After printing, the samples were removed from the substrate using WEDM (wire electrical discharge machining) cutting. Area surface roughness, *S_a_*, was measured using a laser confocal microscope (Keyence VK-X1000, Keyence Corporation, Osaka, Japan). The density of the samples (*ρ*_SLM_) was determined using the Archimedes method, following the ASTM Standard B962 [24]. The relative density (*RD*) of the SLM samples was calculated according to the expression *RD* = *ρ*_SLM_/*ρ*_bulk_, where $\rho_{bulk}=[\left(wt\%TiN\right)\cdot \left(\rho_{TiN}\right)+\left(wt\%AISI420\right)\cdot \left(\rho_{AISI420}\right)]$, and the densities of wrought 420 stainless steel (*ρ*_AISI420_) and Titanium Nitride powder (*ρ*_TiN_) were 7.74 g/cc [7] and 5.21 g/cc [20], respectively. To calculate the deviation errors, each measurement was repeated three times for three different samples.

To prepare the SLM AISI 420 and TiN/AISI 420 specimens for microstructure analysis, the surfaces of the samples were first ground with sandpaper ranging from 80 to 2000 grit. Next, the samples were polished using slurries containing diamond suspensions with particle sizes of 6 µm and 1 µm, followed by cleaning in an ultrasonically agitated alcohol solution. Finally, to reveal the microstructure, the samples were etched using Kroll’s etchant (2 mL HF, 8 mL HNO_3_, and 90 mL DI water) for 5 to 20 s at room temperature, following the recommended protocol [8].

The microstructure of the printed specimens was analyzed using a scanning electron microscope (SEM) equipped with Energy Dispersive Spectroscopy (EDS). The dispersion of TiN particles in the AISI 420 matrix was determined using a transmission electron microscope (TEM) operating at 200 kV. X-ray diffraction (XRD) was employed for phase identification of both the post mixed powders and the two SLM samples, namely AISI 420 and TiN/AISI 420. The XRD analysis was performed using a Bruker D2 Phaser with Cu Kα radiation at 40 kV/40 mA and a wavelength of 1518 nm. The patterns were collected perpendicular to the building direction (BD) at angles ranging from 30 to 90 degrees.

### 2.3. Characterizing Mechanical and Corrosion Properties

To measure the microhardness of the SLM samples, a Vickers microhardness test was performed on a Mitsubishi HV-221 hardness tester with a loading of 5 kg and a dwell time of 10 s. To ensure accurate results, five points were measured at various locations on the top surface of the samples after grinding procedures up to the use of 2000-grit sandpaper, and an average value was calculated.

Moreover, the tensile strength of the samples was measured using a commercial servo hydraulic mechanical testing machine (MTS 810, MTS System Corporation, Eden Prairie, MN, USA). A constant tension load of 50 kN and a strain rate of 1 mm/min were applied during the test. To measure the strain, a uniaxial extensometer was used. The results of the tensile test were calculated based on the average of three experiments.

The electrochemical test was performed in the following steps: Firstly, copper wires were connected to the backside of the samples, and they were embedded in resin. Secondly, the samples were ground with sandpapers ranging from 400 to 4000 grit to obtain a reflective surface, followed by cleaning with ethanol and drying with air. Thirdly, electrochemical measurements were carried out at room temperature, using a corrosive solution of 3.5 wt.% seawater. The experiment utilized three electrodes: the working electrode (post polished SLM samples), the reference electrode (Ag/AgCl), and the counter electrode (platinum sheet). The open-circuit potential (OCP) was recorded for 30 min to ensure the stability of the electrode surface. The scanning rate was set at 1 mV per second, with a voltage range of −300 mV to +1600 mV. Finally, the samples were cleaned with deionized water and isopropyl alcohol before microstructural analysis. Each sample was tested three times to ensure accuracy.

To measure weight loss, epoxy resin was used to mold the SLM cuboids of both AISI 420 and TiN/AISI 420 samples. The four adjacent edges and the bottom edge of each cuboid were sealed with epoxy resin, and the top surface (the polished mirror-like surface) was prepared for etching. The molded samples were then immersed in a 6 wt.% FeCl_3_ solution at 50 ± 5 °C for 48 h. The mass of the sample was measured every 12 h using an electronic balance with an accuracy of 0.001 g. The mass loss per unit area was calculated using the following expression: $m_{loss}=\frac{m_{i}-m_{t}}{s}$ [10], where *m*_i_ was the initial mass of the sample, *m*_t_ was the weight at the measurement moment, and *s* was the corrosive surface area (1 cm^2^). This process was repeated three times with three different samples over 48 h.

## 3. Results and Discussion

### 3.1. Correlation of Relative Density and Surface Roughness

Figure 4a,b illustrates a strong correlation between the processing *VED* and the relative density and surface roughness of the as-printed SLM AISI 420 and TiN/AISI 420 samples. Initially, the two types of samples showed an increase in *RD* with increasing *VED*, followed by a saturation region, and then fluctuations with further increases in *VED*. On the other hand, the surface roughness consistently decreased as the *VED* increased. To fully characterize both *RD* and *S_a_* as a function of *VED*, we identified two transitioning regions: R-I and R-II, which represent insufficient and optimal levels of *VED*, respectively.

Region R-I covered the *VEDs* from 130 to 160 J/mm^3^, and it was characterized by lower density and higher surface roughness. In this region, significant variations in density and roughness were observed. For the AISI 420 samples, the relative density increased from 96.12% (S08) to 98.13% (S06), while the surface roughness decreased from 14.11 µm (S08) to 7.3 µm (S06). Similarly, for the TiN/AISI 420 samples, the relative density increased from 97.53% (S18) to 98.56% (S16), while the surface roughness decreased from 22.34 µm (S18) to 7.53 µm (S16). These results suggest that the surface roughness decreased as the number of main defects, such as lack of fusion porosities, decreased.

In Region R-II, corresponding to the *VED* ranging from 160 to 360 J/mm^3^, the SLM AISI 420 (S01–S05) and TiN/AISI 420 (S11–S15) samples exhibited higher relative density (~99%) and lower surface roughness, indicating a reduced content of defects such as microvoids and pores in the resulting built structures [25]. The surface roughness of the SLM samples decreased with increasing *VED*, reaching a minimum of 3.2 µm for S01 and 4.2 µm for S11. Both the SLM AISI 420 and TiN/AISI 420 samples exhibited slight fluctuations in *RD* and achieved their highest values of 99.62% and 99.22%, respectively, at specific *VEDs*. The AISI 420 reached its maximum *RD* of 99.62% at a *VED* of 205 J/mm^3^, while the TiN/AISI 420 reached its peak *RD* of 99.22% at a *VED* of 286 J/mm^3^. 

Further explanations are provided for the differences in relative density and surface roughness between the SLM AISI 420 and TiN/AISI 420 samples. To achieve the optimal density, the TiN/AISI 420 powder required a VED of 285 J/mm^3^, which was higher than that of 205 J/mm^3^ for the AISI 420 powder. This was due to the reflectivity of TiN, which was 0.68 [26], higher than that of the AISI 420 powder, which was 0.4 [27]. As a result, the absorption effect of TiN/AISI 420 was lower than that of AISI 420 [21], which necessitated the use of more laser energy to improve melt viscosity and wetting characteristics, enhancing densification activity.

Additionally, the surface roughness was observed to be higher for the SLM TiN/AISI 420 samples in comparison to the SLM AISI 420 samples. This difference might be attributed to the presence of TiN within the AISI 420 matrix. During the SLM process, the low density (5.21 g/cc) and high thermal stability of TiN caused it to float to the surface of the molten metal due to buoyancy and Marangoni convection. The presence of the unmelted TiN phase within the melting pool might have caused insufficient flow or a more unstable molten track, resulting in worse surface roughness.

### 3.2. Morphology and Phase Composition

After an extensive analysis of various *VEDs* for fabricating the SLM AISI 420 and TiN/AISI 420 samples, it was determined that the relatively optimal energy densities for printing AISI 420 and TiN/AISI 420 were 205 J/mm^3^ and 285 J/mm^3^, respectively. These *VEDs* were then used to print the subsequent SLM samples of AISI 420 and TiN/AISI 420 to investigate their physical, mechanical, and chemical properties. Referring to Table 2, the SLM AISI 420 and TiN/AISI 420 samples printed using these two *VEDs* are identified as the S04 and S12 samples, respectively. The printed samples based on these VEDs displayed a relative density exceeding 99%. The top-view optical microscope (OM) images of the central part of the S04 and S12 samples are presented in Figure 5a,d, respectively. Their corresponding surface roughnesses were measured to be 4.39 ± 0.50 µm and 4.36 ± 0.48 µm, respectively. Although the roughness values were comparable, the S04 exhibited darker-speckled regions in Figure 5d due to the presence of TiN. The uniform and continuous laser scan tracks along the line scanning direction could be attributed to the good intermixing of various powders in the molten under the appropriate laser processing energy, providing enough time to form a high-density structure during solidification [28].

The high relative density of the as-printed structures, over 99%, indicated a high level of density with a minimal number of pores or voids. To verify this, we examined the polished top and cross-sectional surface morphology of the printed samples. Grinding was utilized to eliminate any potential impact from the surface roughness caused by the SLM process and/or the unevenness produced during the WEDM slicing process. Figure 5b,e shows OM images of the top surfaces of the as-printed S04 and S12 samples, respectively, while their corresponding cross-sectional images are presented in Figure 5c,f. The samples demonstrated smooth and uniform surfaces, with no observable defects such as fusion holes, cracks, or open holes. While tiny pores may appear in the printed structure due to gas entrapment during the SLM process [29], careful adjustment of processing parameters can lead to nearly perfect parts with complete melting and sufficient solidification. The density of the S04 sample was 7.71 kg/m^3^, which was very close to that of wrought 420 stainless steel (7.74 kg/m^3^) and significantly higher than the reported density of 7.67 kg/m^3^ for SLM AISI 420 reported in [7]. This indicates that a higher density is attainable by employing a higher *VED*. The density of the S12 sample was 7.64 kg/m^3^ (*RD* = 99.22%), which was also considerably higher than the reported density of 7.59 kg/m^3^ (*RD* = 98.6%) in [21]. As a result, it is feasible to achieve nearly void-free, high-density SLM AISI 420 and TiN/AISI 420 workpieces using the current SLM processing parameter sets.

Figure 6 displays four XRD spectra: two for the initial powders, AISI 420 and TiN/AISI 420, and the other two for their corresponding SLM samples. Again, the SLM processing was carried out using parameter sets S04 and S12 for AISI 420 and TiN/AISI 420 samples, respectively. The major phases detected in the AISI 420 powder and its SLM sample were martensite (α-Fe) and retained austenite (γ-Fe). The initial TiN/AISI 420 powder and its SLM sample showed traces of the τ-TiN phase, but the peaks were not distinguishable due to the low amount of TiN present. The samples exhibited a well-defined crystalline structure, as indicated by the strongest α-Fe peak, which was caused by the thermal gradient generated during the SLM process [30]. No new phases were observed in the SLM samples compared to the initial powders, implying that no in situ reactions occurred during the SLM processes [17]. In other words, the absence of carbide in the XRD pattern could be attributed to the fast solidification process in the SLM, which prevented the diffusion of alloying elements such as chromium into the matrix. As a result, the carbide formers remained within the iron matrix. Therefore, the solidification microstructures of the SLM samples consisted solely of martensite and retained austenite [31].

The XRD spectra of the AISI 420 powder and its SLM sample were similar, but the martensite phase’s relative intensity was significantly higher in the latter, indicating a higher formation of α-Fe during the SLM process. Notably, comparing the XRD patterns of the initial TiN/AISI 420 powder and its SLM sample showed an even more pronounced increase in α-Fe phase intensity or a decrease in γ-Fe phase intensity. The fast cooling rate during the SLM process likely contributed to the transformation of the retained austenite phase to the martensite phase. The XRD patterns showed that the TiN/AISI 420 powder and the resulting SLM samples contained the τ-TiN phase. This suggests that the TiN particles remained within the AISI 420 matrix throughout the selective laser melting process. 

### 3.3. Microstructures of SLM Samples

To obtain a more detailed observation of the microstructure of the SLM workpieces, we carried out etching on the polished as-printed samples using the procedures outlined in Section 2.2. The resulting top- and side-surface images of the S04 and S12 samples were obtained using the laser confocal microscope and are presented in Figure 7. Figure 7a–c displays the S04 sample, while Figure 7d–f depicts the S12 sample. The top surface morphology of the S04 and S12 samples can be seen in Figure 7a,d, respectively, while the cross-sectional morphology views are illustrated in the other images. Notably, the addition of only 1% TiN into the AISI 420 matrix resulted in a significant difference in surface morphologies.

In Figure 7a,d, a strip-like surface morphology was observed and the strips were roughly distributed in two specific directions with an approximate angle of 67°; this was because a rotating 67° sintering strategy was adopted between layers in the SLM process, as depicted in Figure 2a. The resulting strip width was observed to be around 70 μm. Independent experiments with single-line SLM printing revealed that the built line width was approximately 140 µm. Since the hatch for two consecutive scans was set to be 70 µm, the resulting net path overlap was, thus, approximately 50%. From the side-view images, tracks of the melting during the SLM process, as marked in Figure 7b,e, could be observed. The resulting thickness of each SLM layer was around 50 μm. Because of the jump between the two successive scanning events, curved ends with a spacing of about 70 µm, roughly about the preset hatch distance, were also noticeable. 

Figure 7c,f provides magnified side-viewed images that offer a more detailed view of the microstructures along the building direction. In Figure 7c, the SLM AISI 420 sample (S04) displays the martensite laths and columnar cells, with darker laths indicating the formation of tempered martensite [7]. The AISI 420 powder melted and solidified into a new solid phase due to the use of sufficiently high laser energy during the SLM process, with the lath martensite structure formed in the Kurdjumov–Sachs (K–S) direction. The high cooling rates during the austenite–martensite transformation resulted in the formation of martensite with a specific orientation [29]. In contrast, Figure 7f illustrates small, reconstructed TiN grains that were uniformly dispersed in the S12 sample. These grains originated from the sites of previous austenitic cells or grains that had partially transformed into martensite, which was consistent with the XRD data showing that the peak intensity of the martensite phase is much larger than that of the austenite phase (Figure 6). The high cooling rate during the SLM process resulted in a high nucleation rate and fast dendrite growth, leading to the formation of uniformly dispersed TiN grains. This explains why the printed parts had high hardness. Consequently, the observations suggest that adjusting the laser energy and cooling rate during the SLM process could be used to control the microstructure and properties of the resulting printed workpieces. Specifically, the formation of uniformly dispersed TiN grains can enhance the hardness of the SLM workpieces.

Figure 8a,b presents the SEM images of the etched top surfaces of the S04 and S12 samples, respectively, providing a detailed examination of the microstructure of the SLM workpieces. The rapid melting and solidification processes involved several crystallization processes, which led to the formation of observable micrograins, as shown in Figure 8a. Within each micrograin, several distinct subgrains and laths could be observed, which are consistent with the findings reported in the literature [8,32,33]. The average size of the micrograins and subgrains was estimated to be around 11 ± 3 μm and 1.2 ± 0.4 μm, respectively. The submicron-sized martensite laths within the subgrains were much smaller than those formed by the quenching treatment, as presented by Lu et al. [34]. These results indicate that the current SLM AISI 420 workpieces exhibit excellent mechanical and corrosion properties, which will be discussed in Section 3.5.

In Figure 8b, in addition to micrograins, subgrains, and martensite laths, the presence of TiN particles could also be observed. Due to the participation of TiN particles, the average size of the grain was 7 ± 4 µm, while the subgrain size was increased to 2.5 ± 0.5 µm. The martensite laths were randomly distributed within the subgrains. Based on the XRD spectra analysis in Figure 6, the increased martensite laths were transformed from the austenite phase. Notably, the TiN particles were also able to be evenly and randomly dispersed in the AISI 420 matrix and chiefly tended to distribute along the boundaries of the subgrains. To further observe the distribution of the mixed TiNs in the matrix, EDS compositional analyses were performed on the as-printed S12 samples. Figure 8c shows an enlarged SEM image of the analysis area enclosed by the dotted line in Figure 8b. The main detected elements included Fe, Cr, Ti, N, and C, and the corresponding element maps are shown in Figure 8d–h. Figure 8f,g exhibits consistent positions of Ti and N elements, indicating no TiN bond separation during the SLM process, which was in agreement with the presence of the τ-TiN phase in the XRD analysis discussed in Section 3.2. Although larger TiN particles tended to locate along the grain boundary, the smaller ones were observed inside the grains. The higher processed *VED* used in this study improved the wettability of TiN particles in the melt and reduced the viscosity of the molten pool, resulting in good dispersion of TiN particles in the resulting SLM workpieces. These figures also indicated uniform dispersion of Fe, Cr, C, Ti, and N, without segregation or elemental diffusion. The uniform distribution of metal and nonmetal elements would greatly enhance the strength of metallurgical bonding [14]. Therefore, the incorporation of TiN particles into the AISI 420 matrix could have influenced the size of the micrograins and subgrains in S04 and S12 samples, potentially leading to variations in their mechanical and chemical properties.

Figure 9 provides TEM images and corresponding selected area electron diffraction (SAED) patterns to investigate the presence and interface phases between TiN particles and AISI 420 phases. The dispersed phases within the AISI 420 matrix exhibited clear and strong bonded interfaces, as depicted in Figure 9a,b. These interfaces could contribute to strengthening the structure, playing a crucial role in the mechanical properties. The unique lattice spacings of different phases, including α-Fe (martensitic), γ-Fe (austenitic), and TiN, were observed. The surface of the body cubic structure had a lattice spacing of d (200) = 0.212 nm, which was compared to the standard XRD JCPDS patterns indicating the presence of the BCC phase of the τ-TiN phase. The SAED pattern in Figure 9c confirmed this finding. In addition, the crystal plane spacings of d (111) = 0.407 nm and d (110) = 0.203 nm, along with the SAED patterns shown in Figure 9d,e, corresponded well to the FCC phase of γ-Fe and the BCC phase of α-Fe, respectively. Overall, TEM analysis showed the presence of three main structures, consisting mostly of α-Fe, a portion of γ-Fe, and a small amount of TiN, which were consistent with the XRD results.

### 3.4. Mechanical Properties 

The mechanical properties of the yield strength, modulus of toughness, and hardness of the S04 and S12 samples are presented in Figure 10 and also listed in Table 3. The tensile curve of the S04 sample, Figure 10a, indicates that it underwent elastic deformation up to a yield strength of approximately 725 MPa, followed by plastic deformation, and eventually failed at a stress level of 1270 MPa. The sample had an elongation of 3.82% at failure. In comparison, the S12 sample exhibited improved yield and ultimate tensile stresses of 1260 MPa and 1482 MPa, respectively, representing enhancements of 74% and 17%, respectively, as compared to S04. However, the ductility of S12 was reduced to 2.72%. As a result, the toughness of the S12 sample was calculated to be 31.19 J/m^3^, as shown in the inset of Figure 10a. This value was smaller than that of S04, which was calculated to be 35.96 J/m^3^.

The lower toughness observed in the selective laser melt tensile samples can be attributed to several factors. One potential contributor is the ring-like structure formed during the SLM, which can significantly impact the samples’ toughness. In the SLM process, different locations of the melt pool might have varying grain sizes, resulting in different levels of strengthening due to the grain size effect. Smaller starting grain sizes lead to a more pronounced build-up of dislocations at grain boundaries, increasing the dislocations’ resistance to sliding transfer. Consequently, zones of the melt pool with finer grains are harder to shape when exposed to external forces, leading to uneven deformation and high-stress concentrations that accelerate crack growth. Another factor that potentially contributes to low toughness is the occurrence of martensitic transformation. This phenomenon is commonly associated with microscopic volume expansion, which creates stress at grain boundaries and significant residual stresses among crystalline grains. The resulting stress tends to concentrate at the grain boundaries, increasing brittleness in SLM samples [31]. These factors highlight the need for careful control and optimization of the SLM process to mitigate the formation of defects and improve the mechanical properties of SLM workpieces.

Table 3 compares the mechanical properties of SLM AISI 420 produced in this study with those reported in other studies. The results show that our SLM AISI 420 has superior mechanical properties when compared to previous studies and MIM AISI 420. Specifically, our SLM AISI 420 had a higher tensile strength (UTS), yield strength, and elongation, measuring at 1270 ± 30 MPa, 725 ± 5 MPa, and 3.82 ± 0.3%, respectively, than the UTS of the same SLM AISI 420 in a previous study (UTS: 1050 ± 25 MPa, yield strength: 700 ± 15 MPa, and elongation: 2.5 ± 0.2%) [7] and MIM AISI 420 (UTS: 775 ± 30 MPa, yield strength: 622 ± 15 MPa, and elongation: 1.2 ± 0.3%) [35]. Additionally, our study reported the tensile test results for SLM TiN/AISI 420 for the first time, and the results are promising. The TiN/AISI 420 (S12) printed at *VED* of 285 J/mm^3^ has a higher UTS (1482 ± 30 MPa) and yield strength (1260 ± 5 MPa) than SLM AISI 420, indicating that the addition of TiN to AISI 420 can improve its mechanical properties. However, the elongation (2.72 ± 0.4%) of S12 was lower than that of S04, which is worth noting. It is important to mention that although the properties of both S04 and S12 in this study were lower than those of wrought 420 stainless steel (UTS of 1724 MPa, yield strength of 1482 MPa, and elongation of 8%) [36], our study demonstrated that the SLM process could produce TiN/AISI 420 with improved mechanical properties, making it a viable alternative to traditional manufacturing methods.

**Table 3 materials-16-04198-t003:** Mechanical properties of the SLM AISI 420 and TiN/AISI 420 samples (N.A.—Not Applicable).

Works	Yield Strength (Mpa)	Tensile Strength (MPa)	Elongation, (%)	Hardness, (HV)
Wrought 420 [36]	1482	1724	8	~550 (52 HRC)
MIM 420 [7,35]	622 ± 15	775 ± 30	1.2 ± 0.3	48 ± 2 (HRC)
SLM AISI 420 [7]	700 ± 15	1050 ± 25	2.5 ± 0.2	~600 (55 ± 1 HRC)
SLM AISI 420 [This study]	725 ± 5	1270 ± 30	3.82 ± 0.3	635 ± 15
SLM TiN/AISI 420 [21]	N.A.	N.A.	N.A.	~630 (56.7 HRC)
SLM TiN/AISI 420 [This study]	1260 ± 5	1482 ± 30	2.72 ± 0.4	735 ± 25

Especially noteworthy is the fact that the hardness values obtained for the S04 and S12 samples, which were 635 ± 15 HV and 735 ± 25 HV, respectively, are significantly higher than most values in the literature. Figure 10b displays the reported hardness values of SLM-built samples from AISI 420 powders [6,7,8] and from AISI 420 reinforced with TiC and TiN [14,21] for comparison. The present study achieved a hardness value of 635 HV using a *VED* of 205 J/mm^3^, which was close to the value of 650 HV obtained by Saeide et al. [8] using a *VED* of 173 J/mm^3^, but much higher than the value of 55 HRC (~510 HV) reported by Zhao et al. [6] using a *VED* of 170 J/mm^3^. For S12, the present study achieved a maximum hardness of 735 HV, which was higher than the values for MIM and as-cast AISI 420. 

It is clear that the high hardness of S04 is due to its dense structure (*RD* > 99%) and special microstructure, as outlined in the previous sections. The high relative density means the structure has few defects, leading to strengthening and improved hardness. Additionally, the small grains within the SLM AISI 420 material have many grain boundaries that hinder dislocation movement. The presence of lath martensite within the submicron grain further contributes to the hardening effect. Furthermore, the dense structure and high austenite–martensite transition (as shown in XRD results) strengthen the SLM TiN/AISI 420. The excellent hardness of the SLM TiN/AISI 420 can be fully explained by the retained TiN particles, changes in the matrix microstructure, grain refinement, and the ability of TiN to pin dislocations on austenite grains, as demonstrated in Figure 8c.

### 3.5. Corrosion Characterizations

The corrosion behaviors of the SLM AISI 420 and TiN/AISI 420 were examined by studying their potentiodynamic polarization curves in a 3.5 wt.% NaCl solution. The corrosion potential (*E*), current density (*I*), the anodic slope (*β_a_*), and the cathodic slope (*β_c_*) were determined using the Tafel extrapolation. The resulting values were used to calculate the polarization resistance (*R_p_*) and corrosion rate (*CR*) through the equations $R_{p}=\frac{1}{i}\left(\frac{\beta_{a}\beta_{c}}{\beta_{a}+\beta_{c}}\right)$ and $CR=\frac{i}{\rho A}(k\cdot EW)$ [7]. In the expressions, the density (*ρ*), corrosion surface area (*A*), and equivalent weight (*EW*) of the target were considered, where the values of *EW* were found to be 24.6 and 24.3 for AISI 420 and TiN/AISI 420, respectively. The constant value of *k* used in the calculation was 3.272 m/year and the *A* was fixed at 1 cm^2^. 

The corrosion potential is the point at which the anodic and cathodic reaction rates are equal. Passive layers become more resistant as the corrosion potential increases [34]. In Figure 11a, the corrosion potentials for S04 and S12 are −0.32 V and −0.25 V, respectively. The passivation level is indicated by the corrosion current density, which decreases as the passivation level increases. Table 4 lists all the measured and calculated corrosion parameters for this study. The results show that the corrosion current density of S04 was 1.25 µA/cm^2^, which was higher than that of S12, which was 1.07 µA/cm^2^. As a result, the polarization resistance of S12 was much higher than that of S04, with values of 40.72 kΩ.cm^2^ and 19.35 kΩ.cm^2^, respectively. The resulting corrosion rates for S04 and S12 were 12.2 µm/year and 11.2 µm/year, respectively. The current SLM AISI 420 demonstrates a significant improvement in corrosion resistance compared to both the wrought cast 420 stainless steel [37] and the SLM AISI 420 [7]. To further validate the obtained corrosion rates, a weight loss test was conducted by immersing both S04 and S12 samples in a 6% w.t FeCl_3_ solution at 50 °C for 48 h. The weight loss of S04 was found to be 5.1 mg/mm^2^, while that of S12 was 4.66 mg/mm^2^, as shown in Figure 11b. These results were consistent with the electrochemical measurements and confirm that S12 exhibited better corrosion resistance than S04. Therefore, S12 has the potential to be used in applications that require high corrosion resistance.

The OM images of S04 and S12 surfaces after corrosion tests in the NaCl solution are shown in Figure 12. The images revealed the presence of pores on the surface, and the rust formation around the holes indicates pitting corrosion [37]. The corrosion resistance capability of the samples can be determined by counting the number and size of pits. In particular, Figure 12a,b depicts the images for S04, which exhibited more pits and larger and deeper pit sizes than those shown in Figure 12c,d for S12. This observation provided further evidence that S12 has higher corrosion resistance than S04.

Figure 13 depicts SEM images of the corroded surfaces of samples S04 and S12, which were immersed in a FeCl_3_ solution at 50 °C for 48 h. The two samples exhibited distinct surface morphologies, suggesting that the corrosion progressed through different mechanisms. In S04 (Figure 13a), the inset image clearly shows grain boundaries, and corrosion mainly occurred inside the grains. In contrast, S12 (Figure 13b) exhibits a dendritic structure after corrosion, and the regions around the grain boundaries were etched away and not as visible. The observed differences in corrosion behavior could be attributed to the microstructural features of the samples, which contain various phases and nanoparticles (TiNs, Figure 8c). The austenite phase is known to provide higher corrosion resistance than the martensite phase [38]. In SLM TiN/AISI 420, the presence of TiN around the grain boundaries resulted in a lower proportion of austenite than martensite due to the martensite–austenite transformation (Section 3.2), leading to a reduction in Cr atoms at the boundary and sensitization to intergranular corrosion [39]. As shown in the SEM images, the SLM AISI 420 sample exhibits a cellular structure, while the SLM TiN/AISI 420 sample exhibits a martensite lath after corrosion.

## 4. Conclusions

This study investigated the influence of laser volume energy density (*VED*) on the characteristics of AISI 420 stainless steel and TiN/AISI 420 composite fabricated via the selective laser melting (SLM) process. The composite was composed of 1 wt.% TiN and the average particle sizes of AISI 420 and TiN powders used were 45 µm and 1 µm, respectively. A new two-stage mixing method was proposed to produce the TiN/AISI 420 composite powder for SLM. The morphology, mechanical, and corrosion properties of the specimens were analyzed, and their correlations with microstructures were investigated. The following key findings were identified: The two-stage mixing method demonstrated that the finer TiN particles were able to be evenly coated on the surfaces of the AISI 420 powder, while the larger TiN particles were uniformly dispersed throughout the AISI 420 powder.The surface roughness (*S_a_*) and relative density (*RD*) of the SLM AISI 420 and TiN/AISI 420 specimens were dependent on the processing *VED*. The surface roughness decreased with an increase in *VED*. At the *VED* of 360 J/mm^3^, the *S_a_* of SLM AISI 420 and TiN/AISI 420 specimens were at 3.2 µm and 4.2 µm, respectively. The stability of the melting pool under higher *VED* was the reason for the low surface roughness.The highest *RDs* of the SLM AISI 420 and TiN/AISI composite were 99.62% and 99.22%, respectively, and occurred at the *VEDs* of 205 and 285 J/mm^3^, respectively. The SLM process generated no significant defects if appropriate laser processing *VED* and scanning strategy were employed. As a result, a near-full-density SLM workpiece was able to be obtainable.SLM AISI 420 samples had martensite and retained austenite as the main phases, while SLM TiN/AISI 420 had TiN, martensite, and retained austenite as the main phases. A ring-like structure of micrograins was formed by retained austenite in the grain boundary and submicron martensite lath in the grain, leading to high performance in hardness and corrosion resistance.TiN particles dispersed uniformly into the AISI 420 matrix, acting as a reinforced phase and as a nuclear site for finer grain formation at a *VED* of 285 J/mm^3^. As a result, a hardness up to 735 HV was obtained, which was higher than any available reported results.The corrosion test demonstrated that corrosion occurred inside the micrograin of SLM AISI 420 stainless steel, whereas it happened mainly along the grain boundary for SLM TiN/AISI 420 composite due to differences in their resulting microstructures. The SLM TiN/AISI 420 composite exhibited good corrosion resistance in 3.5 wt.% NaCl and 6 wt.% FeCl_3_ solutions compared to the SLM AISI 420.

Consequently, this study demonstrates the potential for producing high-quality SLM AISI 420 and TiN/AISI 420 composites, making them promising candidates for various manufacturing applications. However, further investigations are needed to fully understand the composite’s potential, including exploring a wider range of processing parameters and their influence on microstructure and material properties. Additionally, studying the effects of varying TiN content and heat treatment is crucial for determining optimal composition and processing conditions to enhance overall performance. Incorporating these suggestions into future research will enable the successful application of the TiN/AISI 420 composite in various industries.

## Figures and Tables

**Figure 1 materials-16-04198-f001:**
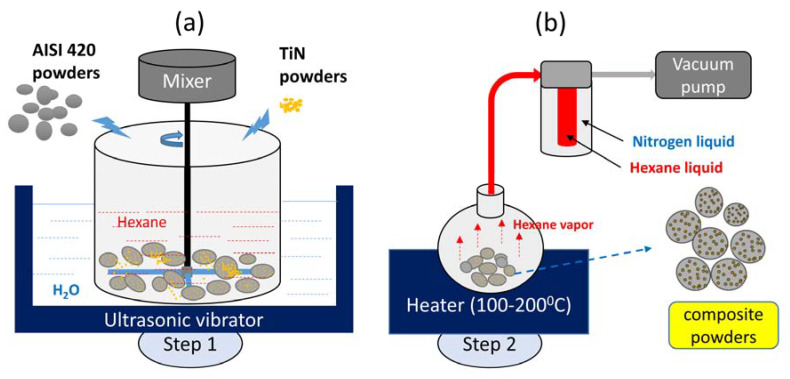
Illustrates the two steps involved in the first stage of the two-stage mixing process used in this study. In Step 1, AISI 420 and TiN powders were mixed under hexane solvent using a propeller mixer and an ultrasonic vibrator. In Step 2, the solid–liquid mixture was dried using a vacuum line system and heater to remove the hexane.

**Figure 2 materials-16-04198-f002:**
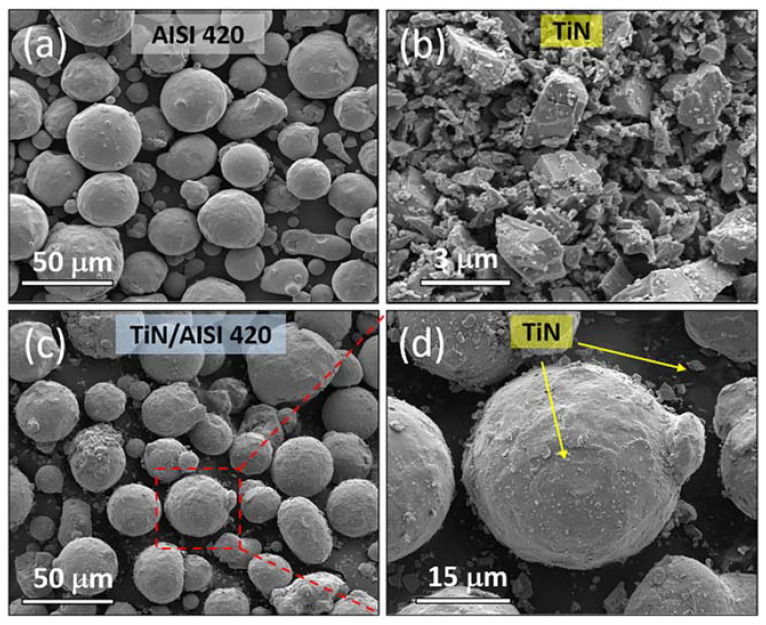
SEM images depicting the morphologies of the powders used in this study: (**a**) As-purchased AISI 420 powder; (**b**) As-purchased TiN powder; (**c**) Mixed TiN/AISI 420 powder; (**d**) Magnified image showing finer TiN particles coated on the surfaces of the AISI 420 powder and larger TiN particles dispersed throughout the AISI 420 powder.

**Figure 3 materials-16-04198-f003:**
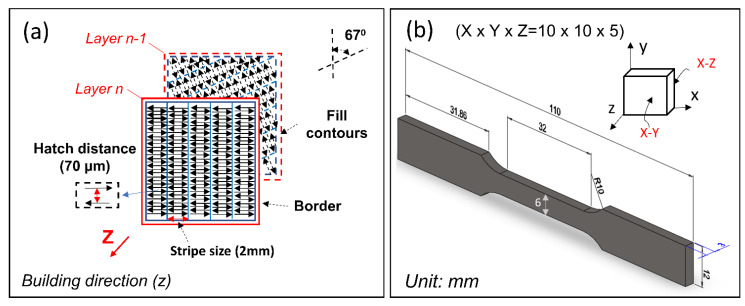
Illustrates the scanning strategies for (**a**) and the dimensions of the cuboid sample used for property characterizations and the dog-bone sample used for the tensile strength test (**b**). The building direction was along the *z*-axis.

**Figure 4 materials-16-04198-f004:**
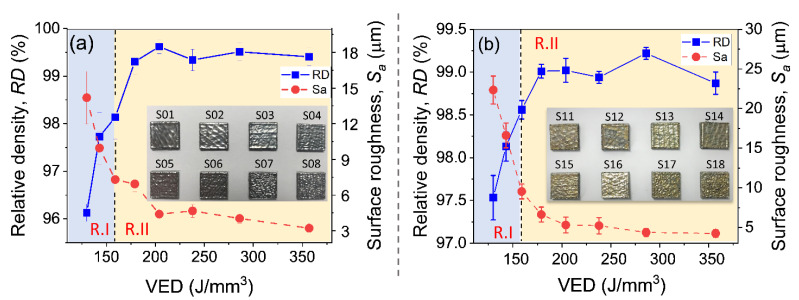
Illustrates the relationship between relative density (RD) and surface roughness (Sa) concerning processing laser volume energy density (VED) divided into Region I (R.I—VED of 130–160 J/mm^3^) and Region 2 (R.II—VED of 160–360 J/mm^3^) for the as-printed AISI 420 (**a**) and TiN/AISI 420 (**b**) samples. The insets display the studied samples’ surface optical microscopy (OM) images (S01–S08 for the SLM AISI 420s and S11–S18 for the SLM TiN/AISI 420s).

**Figure 5 materials-16-04198-f005:**
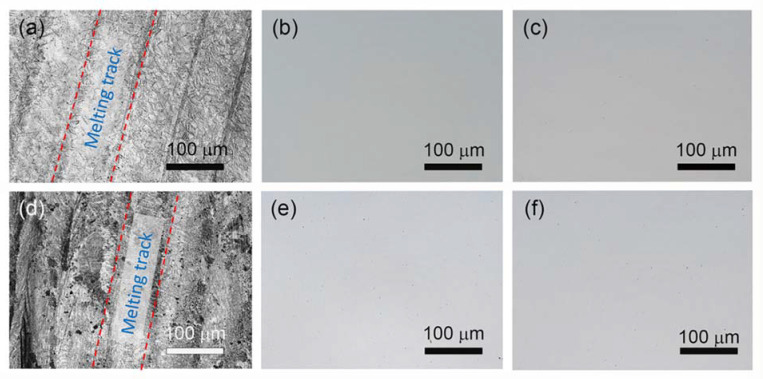
Optical microscope images of SLM AISI 420 samples for (**a**–**c**) and SLM TiN/AISI 420 samples for (**d**–**f**), fabricated using the processing parameters of S04 and S12, respectively. Panels (**a**,**d**) show top–view surface images (X–Y plane), the red dashed lines indicate the melting track area, while panels (**b**,**e**) show the corresponding post-polished top-view surface images. Panels (**c**,**f**) display the post-polished side-view surface images (X–Z plane).

**Figure 6 materials-16-04198-f006:**
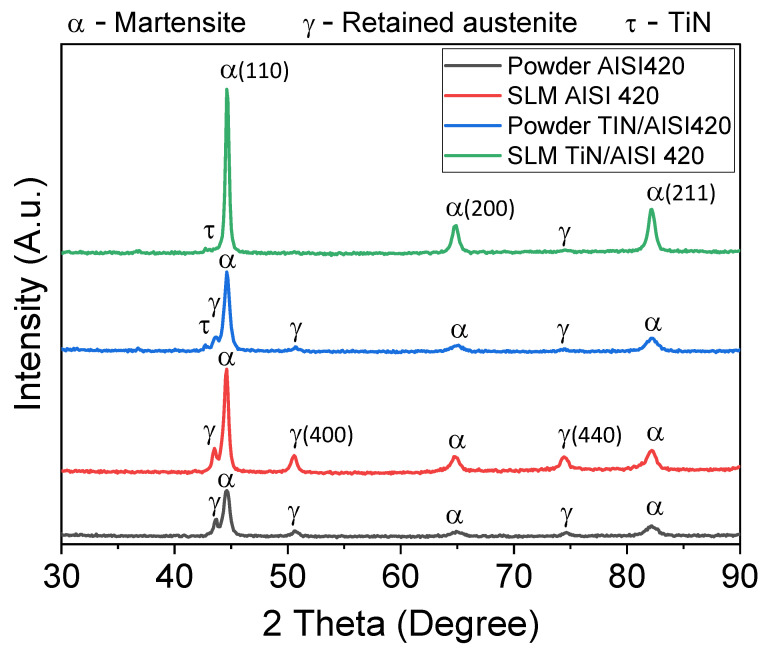
XRD spectra of powders and SLM samples for both AISI 420 and TiN/AISI 420. The SLM samples were manufactured using processing parameters of S04 and S12, respectively.

**Figure 7 materials-16-04198-f007:**
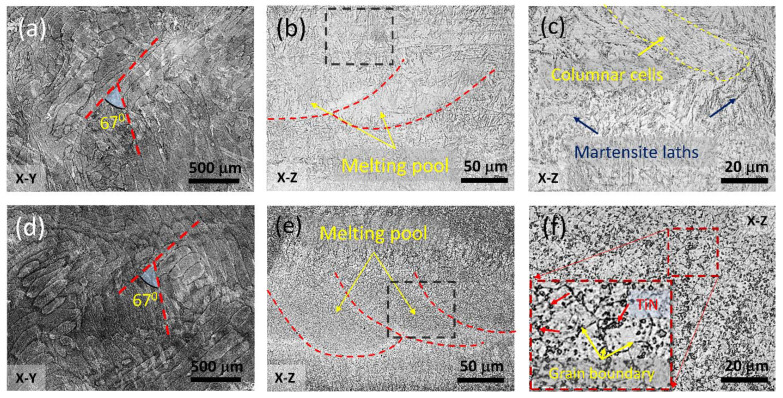
Laser confocal microscope images of the etched surface morphologies of SLM AISI 420 (S04) and TiN/AISI 420 (S12) samples. (**a**–**c**) are for S04, while (**d**–**f**) are for S12. In the building layer surface along the X–Y plane, (**a**,**d**) show observable tracks of the scanning strip. (**b**,**e**) indicate distinguishable marks of the molten pools on the cross-sectional X–Z plane. Panels (**c**,**f**) are magnified images from the black box area of (**b**,**e**), respectively. Columnar cells and martensite laths are visible in (**c**). The inset in (**f**) displays the TiN particles and grain boundaries from the magnification area of the red box area.

**Figure 8 materials-16-04198-f008:**
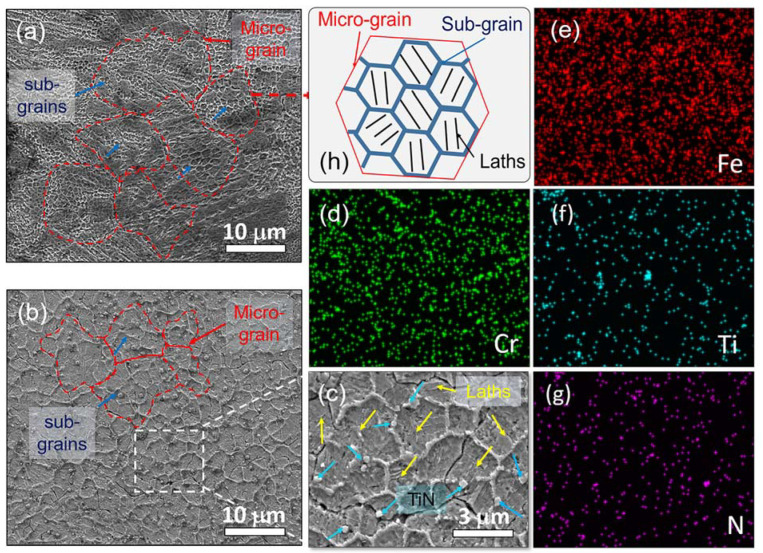
SEM images of the metallography of the SLM S04, (**a**), and S12, (**b**), samples, where the ring-like structure consisting of micrograins and subgrains are visible. (**c**) displays a magnified image of the enclosed area (white box) in (**b**), where TiN particles (indicated by blue arrows) were found to be located along the grain boundaries and also dispersed inside the grains. Martensite laths (pointed out by yellow arrows) distributed inside the grains can also be observed. (**d**–**g**) show the images of EDS element mappings of Cr, Fe, Ti, and N, respectively. (**h**) schematically illustrates the micrograin, subgrain, and lath structures inside the grains.

**Figure 9 materials-16-04198-f009:**
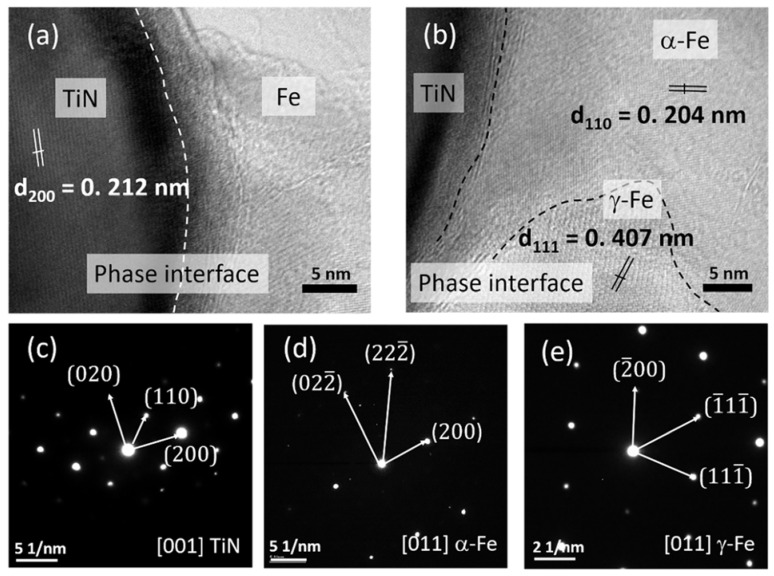
TEM images of the S12 sample: (**a**) The phase interface between reinforced TiN and matrix Fe; (**b**) TiN, martensitic, and austenitic phases are identifiable; (**c**) Selected area electron diffraction (SAED) images of the TiN phase from (**a**,**d**,**e**), respectively, show the SAED images of α-Fe and γ-Fe phases from (**b**).

**Figure 10 materials-16-04198-f010:**
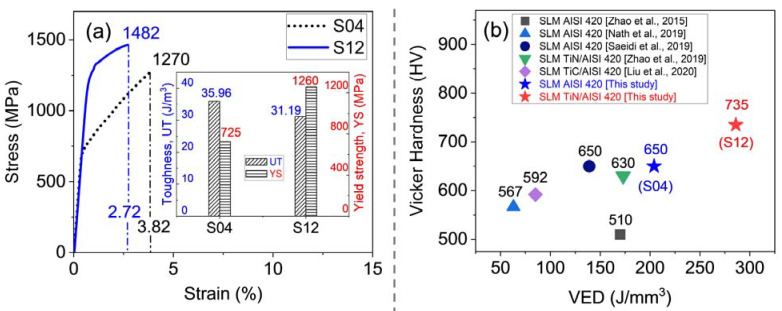
The mechanical properties of the SLM samples: (**a**) The stress–strain curves of S04 and S12 are shown, with the inset displaying the toughness and yield strength for both samples; (**b**) The hardness values of SLM AISI 420 [6,7,8], TiN-reinforced AISI 420 [21], and TiC-reinforced AISI 420 [14] are compared for benchmarking purposes.

**Figure 11 materials-16-04198-f011:**
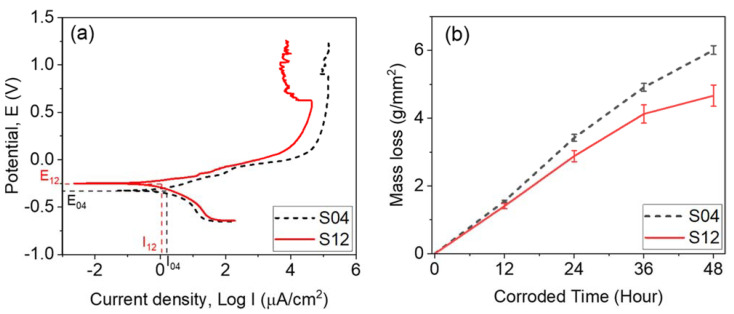
Illustrates the corrosion resistance characterization results of the S04 and S12 samples. In (**a**), the polarization curves in 3.5 wt.% NaCl are shown. In (**b**), the weight loss test results in 6 wt.% FeCl_3_ solution are presented.

**Figure 12 materials-16-04198-f012:**
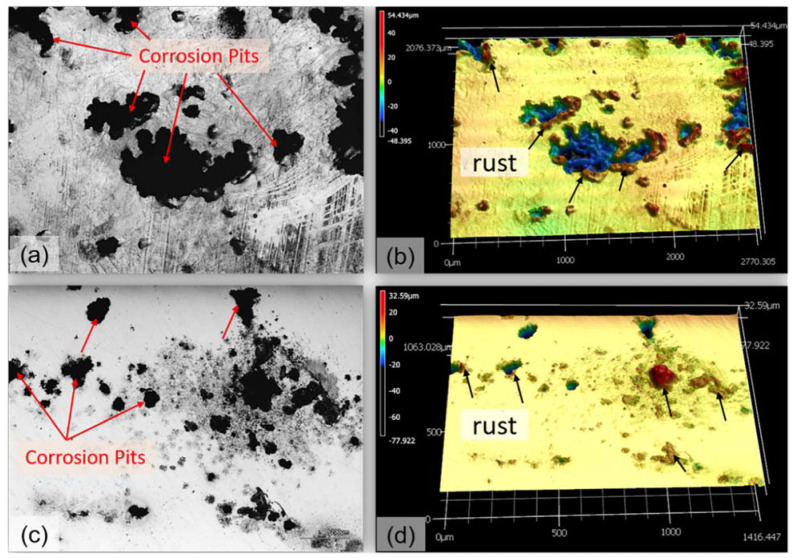
(**a**,**c**) show the optical microscope (OM) images of corroded surface morphologies of S04 and S12 samples, respectively. (**b**,**d**) represent their corresponding 3D images. The corrosion was executed in a 3.5 wt.% NaCl solution.

**Figure 13 materials-16-04198-f013:**
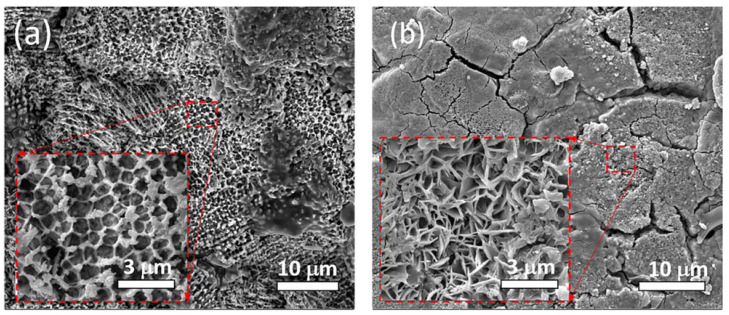
(**a**,**b**) present SEM images of the corroded surface morphologies of the S04 and S12 samples, respectively. The insets depict magnified images of the enclosed areas. The corrosion was performed in a 6 wt.% FeCl_3_ solution.

**Table 1 materials-16-04198-t001:** Chemical compositions of as-purchased AISI 420 and TiN powders (weight percent, wt.%).

Powder	Ti	N	O	C	Ni	Mn	Si	Cr	Fe
AISI 420	-	-	-	0.33	0.06	0.34	0.49	13.09	Bal.
TiN	Bal.	22.3	1.1	0.11	0.03	0.01	-	-	-

**Table 2 materials-16-04198-t002:** The SLM process parameters used for manufacturing AISI 420 and TiN/AISI 420 samples.

SLM Parameters	Value							
Laser power, P (W)	250							
Layer thickness, t (mm)	0.05							
Hatch distance, h (mm)	0.07							
Scanning speed, v (mm/s)	200	250	300	350	400	450	500	550
Volume energy density, VED (J/mm^3^)	360	285	240	205	180	160	145	130
Remark AISI 420 samples	S01	S02	S03	S04	S05	S06	S07	S08
Remark TiN/AISI 420 samples	S11	S12	S13	S14	S15	S16	S17	S18

**Table 4 materials-16-04198-t004:** Corrosion behavior of SLM AISI 420 and TiN/AISI 420 in 3.5 wt.% NaCl solution.

Sample	E (V)	i (µA/cm^2^)	R_p_ (kΩ.cm^2^)	CR (µm/year)
Wrought [37]	−0.40 ± 0.02	2.1 ± 0.1	18.7 ± 0.35	23 ± 2
SLM AISI 420 [7]	−0.39 ± 0.03	2.85 ± 0.4	17.1 ± 0.52	28 ± 2
SLM AISI 420 (This study)	−0.32 ± 0.01	1.25 ± 0.1	22.2 ± 1.78	13 ± 1
SLM TiN/AISI 420 (This study)	−0.25 ± 0.01	1.07 ± 0.1	24.9 ± 2.34	11 ± 1

## Data Availability

The data presented in this study are available on request from the corresponding author.

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
