# Peer review of "Enhancing Mechanical and Corrosion Properties of AISI 420 with Titanium-Nitride Reinforcement through High-Power-Density Selective Laser Melting Using Two-Stage Mixed TiN/AISI 420 Powder"

_materials, 2023, doi:10.3390/ma16114198_

Round 1

Reviewer 1 Report

The author has well demonstrated the mechanical and corrosion properties of AISI 420 with titanium Nitride Reinforcement through High-power-density selective laser melting using Two-stage Mixed TiN/AISI 420 Powder.

·       The reviewer is interested to know what is specific advantages of two-stage mixing method over existing methods.

·       It is difficult to clearly visualize the microscopical changes in Figure 7 and seems to be a poor-quality image from the redear point of view. The author is suggested to use the good quality microscopic image to reveals the melting pool, columnar cell and grain boundaries.

·       The reviewer is interested to know on what basis the author has selected SLM process parameters such as Laser power, Layer thickness and Hatch distance. Whether those parameters influence the mechanical and corrosion properties of selective laser melted AISI 420 with TiN reinforced materials or not?

·       The author is strongly advised to mention the limitation of the present study in the conclusion section.

·       The author is strongly advised to mention the future scope of the present research work in the conclusion section.

Quality of English is seems to be fine. However, the author is suggested to take careful revision of entire manuscript to avoid any typos and grammatical errors while submitting the revised manuscript.

Author Response

Dear Referee,

The authors would like to express their sincere appreciation for your valuable assistance in providing constructive comments during the review of our manuscript. Your feedback has been instrumental in improving the quality of our work, and we are truly grateful for your time and effort.

In the revised manuscript, all the suggested corrections have been diligently addressed and are clearly marked in red.

Once again, we extend our heartfelt thanks for your invaluable contribution.

Best regards, 

Jeng-Rong Ho

Reviewer 2 Report

The current research article pertains to a very interesting topic, since a study regarding the utilization of a two-stage mixing method for the preparation of the powder material before the SLM process has been caried out. In general, it is well written, with an adequate literature review, while the followed methodology is also clearly presented. Moreover, the obtained results are also interesting and with practical value as well, whilst an extensive analysis and discussion is also included. Only some minor improvements are suggested:

More specifically:

the quality of Figures 3, 4, 7 and 10 needs an improvement.

was a standard followed for the surface roughness measurements? Please provide more details (e.g. cut-off length, number of measurements, etc.).

are the dog-bones samples dimensions according to a standard?

• authors defined the optimal VEDs for AISI 420 and TiN/AISI 420 material. Do authors consider that the same VED but with different process parameters combination (i.e., different combination of laser power and scanning speed) will also resulted optimal properties. Please comment on this based on your expertise as it would be definitely interesting.

based on authors expertise, would a post process heat treatment improve the material properties? For example, would a higher elongation would be achoeved for the TiN/AISI 420?

Considering the aforementioned, in my opinion, the current paper can be accepted after a minor revision

Minor editing of English language is required

Author Response

(The authors gave the same response as above.)
